# In the Right Place at the Right Time: miRNAs as Key Regulators in Developing Axons

**DOI:** 10.3390/ijms21228726

**Published:** 2020-11-18

**Authors:** Eloina Corradi, Marie-Laure Baudet

**Affiliations:** Department of Cellular, Computational and Integrative Biology—CIBIO, University of Trento, 38100 Trento, Italy; eloina.corradi@unitn.it

**Keywords:** axonal compartment, local translation, miRNA, miRNA trafficking, miRNA localization, neural development, neuronal circuit formation

## Abstract

During neuronal circuit formation, axons progressively develop into a presynaptic compartment aided by extracellular signals. Axons display a remarkably high degree of autonomy supported in part by a local translation machinery that permits the subcellular production of proteins required for their development. Here, we review the latest findings showing that microRNAs (miRNAs) are critical regulators of this machinery, orchestrating the spatiotemporal regulation of local translation in response to cues. We first survey the current efforts toward unraveling the axonal miRNA repertoire through miRNA profiling, and we reveal the presence of a putative axonal miRNA signature. We also provide an overview of the molecular underpinnings of miRNA action. Our review of the available experimental evidence delineates two broad paradigms: cue-induced relief of miRNA-mediated inhibition, leading to bursts of protein translation, and cue-induced miRNA activation, which results in reduced protein production. Overall, this review highlights how a decade of intense investigation has led to a new appreciation of miRNAs as key elements of the local translation regulatory network controlling axon development.

## 1. Introduction

During neuronal circuit assembly, the distal tip of the growing axon navigates a complex environment and ultimately reaches its target where it branches and establishes connections with the synaptic partners. Each of these phases of axon development is largely controlled by extracellular signals that induce local cytoskeletal remodeling. As the axon elongates, its tip becomes distant from the cell body and can no longer rely on the soma to promptly supply the proteins that it needs for its development. To overcome this challenge, the axon progressively acquires a high degree of autonomy that enables it to independently manufacture its building blocks. Part of this autonomy is conferred by the delocalization of the translational machinery to the axon to fuel their development through the localized production of proteins.

The molecular mechanisms of mRNA translation in axon development have been clarified through decades of investigation [1,2,3]. Initial studies have documented that selected guidance cues trigger the translation of specific axonal mRNAs within minutes, providing insight into the temporal regulation of mRNA translation and leading to the concept of local translation on demand [4,5,6,7,8]. The advent of large-scale transcriptomics, translatomics, and proteomics analyses have revealed a hitherto unsuspected rich, complex and stage-specific population of axonal transcripts [9,10]. It was confirmed that specific transcripts are selected for translation and that unique protein signatures are synthesized within axons in response to extracellular cues and, unsuspectedly, under basal conditions [11,12]. Moreover, intra-axonal translational events have been observed in adult neurons [11,13]. Thus, beyond the initial evidence of the important role of local translation during neuronal development, such a mechanism appears to be important throughout the neuron’s lifespan, including adulthood [14,15]. Additionally, aspects of the mechanisms underlying the spatial restriction of mRNA translation are starting to be clarified. While it was known that mRNA translation is triggered asymmetrically within the growth cone in relation to the side of cue exposure [5,6], recent studies revealed additional focal hotspots of de novo protein synthesis appear at sites of branch emergence, on RAB7A endosomes, and in association with mitochondria [16,17,18]. Studies from the past 30 years concur to show that the axonal translation machinery triggers the production of the right protein, at the right time, and at the right place to promote axon development. This begs the question of how this process is so exquisitely regulated.

The molecular nature of regulators of axonal mRNA translation and their mechanisms of action are beginning to be elucidated. In general, *trans* regulators of mRNA translation fall within the two broad categories of RNA binding proteins (RBPs) and non-coding RNAs (ncRNAs), the latter category being overwhelmingly constituted by miRNAs. MiRNAs constitute a large category of short, non-coding RNAs that play widespread regulatory roles in post-transcriptional steps of gene expression, mainly affecting mRNA stability and translation [19]. They act essentially as specificity determinants for effector proteins, which are represented by Argonautes (AGOs) in eukaryotes. MiRNAs act through imperfect matches to their target sequences [20], and thereby constitute in vertebrates a vast level of gene expression regulation. In principle, miRNAs are capable of affecting virtually all transcripts within the cell. The scope of miRNA-mediated post-transcriptional gene regulation makes them particularly suited for the spatiotemporal and selective regulation of mRNA translation within axons.

The hunt for regulators of axonal translation has been increasingly zeroing in on the role of miRNAs. Research performed mostly in the last decade has produced a seizable body of evidence showing that miRNAs significantly contribute to support the functional autonomy of the developing axons by exquisitely regulating mRNA translation locally into newly synthesized proteins.

In this review, we will detail the molecular mechanisms employed by miRNAs to regulate axon development during neuronal circuit formation. We will first compare the numerous studies that have revealed the presence of a rich and diverse repertoire of miRNAs within the developing axons. We will also present the recent advances on the mechanisms of miRNA transport to the tip of the axon. We will finally explore how the axon critically exploits miRNAs to ensure that a subset of mRNAs are selected for translation locally on demand into proteins that promotes its outgrowth, steering, and branching.

## 2. In the Right Place: miRNAs Are Translocated to Developing Axons

The asymmetrical distribution of mRNAs within cells play important roles in cell fate [21,22], polarization [23,24], cell motility [25,26,27], oogenesis, and embryonic development [28,29,30]. While the molecular mechanisms of mRNA spatial distributions are well-studied, those of ncRNAs are still largely unknown. Emerging evidence indicates that miRNAs specifically localize within the brain, brain regions, and brain cells [31,32,33,34,35,36,37], but also in the subcellular compartments, including the axon [38,39,40,41,42,43,44,45]. Here, we briefly summarize the current knowledge of miRNA spatial distribution and associated transport mechanism within the axon.

### 2.1. miRNA and pre-miRNA Detection in Axons

MiRNA profiling within axons of different cell types from different species has revealed the presence of a rich and complex repertoire of axonal miRNAs (Table 1) [38,39,40,41,42,46]. Specific miRNAs also appear enriched in axons compared to the soma [38,39,40,46], but the identity of these molecules varies between studies (Table 1). No signature appears to emerge, and miRNAs enriched in a given axon type can be depleted in another (e.g., miR-181a-1-3p and miR-361 are enriched in axons of mice cortical neurons [39], but depleted in axons of mice motor neurons [46]). These discrepancies might be due to differences in neuronal types or age (i.e., developmental status vs. adulthood).

Nevertheless, these data suggest that miRNAs exert an important and conserved regulatory role to shape the local proteome in this compartment, and by extension, in neural circuit formation (Table 1) as confirmed by several studies (see below).

Is an axonal miRNA signature emerging from these studies? Our survey of the available datasets yields interesting, if not always consistent clues. Subsets of miRNAs are specifically expressed in the brain (miR-103, -107, -124, -129, -134, -181, -191, -218) [34] or in brain and nervous system (NS)-related tissues (miR-9-5p, -124-3p, -219a-5p, -338-3p) [33], while seven unique miRNAs appear to be consistently detected in all axon types (miR-17-5p, miR-20a, miR-30b, miR-30c, miR-125a-5p, miR-125b-5p, miR-204) (Figure 1a). The relatively small size of the overlapping axonal miRNA pool is likely to stem from the wide differences in biological systems and detection techniques among the studies. While microarray and RT-qPCR array reveal a predefined miRNA set [38,39,40], next-generation sequencing (NGS) ensures an unbiased detection of the entire miRNA population [41,42]. Interestingly, when we separately compare arrays and NGS-derived datasets, a larger proportion of miRNAs appears to be shared regardless of the cell type under investigation (Figure 1b,c). In particular, approximately 40% of axonal miRNAs are shared between the two available NGS-generated datasets (Figure 1c). Overall, while the paucity of published directly comparable datasets does not allow a firm conclusion, these studies support the possibility that axons might indeed express a bona fide miRNA signature. Additional miRNA sequencing analyses from different species and neuron types will help clarify whether such a signature really exists.

Intriguingly, not only mature miRNAs but also precursor miRNAs (pre-miRNAs) are detected in axons. miRNAs are produced in the cells through sequential steps. They are first transcribed as primary miRNAs (pri-miRNAs), processed by Drosha into precursor molecules (pre-miRNAs) in the nucleus, exported to the cytoplasm, and further cleaved by Dicer to obtain the active mature molecule (miRNA). Therefore, pre-miRNAs can be considered as an inactive form of miRNA. The presence of pre-miRNAs in axons has been confirmed through a candidate (RT-qPCR) and a high-throughput approach (NGS) (Table 2).

### 2.2. Axonal miRNA Transport

The detection of mature and precursor miRNAs in axons suggests the existence of a specific transport mechanism to translocate these molecules to subcellular regions. While mRNA trafficking mechanisms are well described in different subcellular models [48,49,50], how ncRNAs, and in particular miRNAs, are directed to cellular compartments is just starting to emerge. In principle, at least three models for miRNA neuronal transport can be envisioned: (i) miRNAs translocate toward their destination by hitchhiking on their target mRNAs, (ii) miRNAs are transported independently from their target loaded onto AGO2 or (iii) as inactive pre-miRNAs.

#### 2.2.1. Axonal miRNA Transport: Mature miRNAs

Due to the technical challenges associated with tracking endogenous short sequences, the first two models have not yet been experimentally verified. Exogenous cy3-tagged miR-124 was found to be transported with lysotracker-marked acidic compartments in axons [51]. In addition, an RBP-mediated axonal transport of miRNA associated with its targets has been proposed [52]. Wang and colleagues [52] have produced several lines of evidence supporting a role for FMRP (Fragile X mental retardation protein) in mediating axonal delivery of *Map1b* (Microtubule-associated protein 1B) and *Calm1* (Calmodulin 1) transcripts in rodent dorsal root ganglion (DRG) axons. FMRP regulates mRNA transport and translation [53] and has been suggested to associate to RNA induced-silencing complex (RISC) [54]. In FMRP-deficient DRG neurons, the axonal trafficking of exogenous *Map1b* and *Calm1* 3’UTR (untranslated region) is reduced, and both the transcripts and protein level of MAP1B and Calmodulin decrease specifically in axons and not in the cell body. These effects are likely to be direct, as FMRP associates and co-localizes with miR-181d, *Map1B*, and/or *Calm1* as shown, respectively, by immunoprecipitation in whole DRG neurons and by immunostaining/in situ hybridization (ISH) in axons. Target protectors are antisense oligonucleotides which mask the miRNA-responsive element (MRE) within a target to which miRNA hybridize for mRNA-specific silencing [20,55]. Using target protectors, it has been possible to uncouple miR-181d from *Map1b* and *Calm1* mRNA, resulting in increased levels of both proteins within axons. Overall, this study suggests an axonal repressive interaction of miR-181d with its mRNA targets via FMRP [52]. However, whether this interaction mediates the miRNA axonal transport itself, alone, or in combination with its targets remains to be elucidated.

#### 2.2.2. Axonal miRNA Transport: pre-miRNAs

Evidence in support of the third model, or that miRNAs are transported in dendrites and axons in their precursor, inactive form (pre-miRNA), is increasingly becoming available (Figure 2). The first such evidence came from the observation by the Schratt’s group that exogenous fluorescently labeled pre-miR-134 accumulates in dendrites of rat hippocampal neurons through its direct association with the RBP DEAH-box helicase DHX36 [56], and that such localization is affected by neuronal activity, in particular by increased levels of brain-derived neurotrophic factor (BDNF) [57]. Additional clues of a pre-miRNA transport mechanism have come from studies carried out in axons. Exogenous pre-miR-338 localizes in distal axons in a microtubule-dependent manner. This precursor has been found to be associated with mitochondria, by RT-qPCR from axonal mitochondrial fractions, and with specific proteins (VDAC1, KIF5A, and YB1) by pre-miRNA pulldown followed by Western blotting within cultured axons of rat superior cervical ganglion neurons (SCGs) [44].

It is only recently that trafficking dynamics of pre-miRNAs in neurons was finally revealed thanks to the development of adequate tools to track endogenous molecules [45]. One of the critical limitations of studying endogenous pre-miRNA (and miRNAs) transport is their extremely short length (pre-miRNA ≈60 nt and mature miRNAs only ≈21–23 nt). Molecular beacon (MB) technology is a powerful method that overcomes these limitations because it can be designed to specifically hybridize to short single-stranded nucleic acid sequences. MBs exist in a closed stem-loop non-fluorescing conformation that opens, emitting fluorescence, upon hybridization with its RNA target sequence. Using MB and a combination of live and super resolution imaging approaches, we discovered that endogenous pre-miR-181a-1 is actively transported bidirectionally along axons while docked to late endosome/lysosomes (LE/Ly) [45]. Pre-miR-181a-1 is stored within the growth cone central domain [45]. It might seem somewhat surprising that pre-miRNAs adopt an organelle-mediated transport, which is different from the more established trafficking mechanisms of mRNAs. mRNAs travel packed into the ribonucleoprotein (RNP) complex [58], and these RNP granules associate directly or indirectly through adaptors to motor proteins that ensure translocation along cytoskeletal tracks [59,60,61,62]. Yet the transport of RNA associated with membranous organelles is not unprecedented: non-canonical endosome-based mRNA trafficking is established in fungi [63,64], and a lysosome-mediated RNA granule transport was also recently documented in axons [65]. Furthermore in non-neuronal cells, mature miRNAs, miRNA-repressible mRNAs, and components of the miRNA processing machinery associate with LE/Ly [66,67]. Thus, endosome-mediated RNA transport might be a hitherto overlooked common translocation mechanism in eukaryotic cells. Since derailed transport has been linked to several neurological diseases [68], it would be of great interest to investigate whether impaired endosome-mediated pre-miRNA trafficking is somehow implicated in these pathologies.

Profiling analyses have revealed a rich repertoire of miRNAs in diverse populations of developing axons and have supported the possibility of a distinct axonal miRNA signature. The abundance and diversity of axonal miRNAs suggest that they play important roles in this compartment.

## 3. At the Right Time: Axonal miRNA Regulate mRNA Translation on Demand

Experimental evidence points to a crucial role of axonal miRNAs in regulating the temporal expression of newly synthesized proteins. Two key modes of actions are starting to emerge both involving cue-induced modulation of axonal miRNA activity. In the first one, miRNAs silence mRNA translation constitutively, and axonal stimulation lifts this repression, thereby allowing the generation of newly synthesized proteins. In the second scenario, miRNAs are present within axons but in an inactive form. Cues trigger miRNA activation and the concomitant silencing of axonal mRNAs. These two mechanisms impact axon development at different stages (Figure 3).

The investigation of intra-axonal miRNAs mechanisms is technically challenging, and several strategies have been developed to focus specifically into the axonal subcompartment and to uncover the miRNA-mediated molecular mechanisms (Figure 4). The detailed miRNA-mediated mechanisms and their effects on neuronal behavior are presented below.

### 3.1. Axonal Stimulation Releases miRNAs-Mediated mRNA Silencing

#### 3.1.1. Initial Evidence

Initial studies have provided evidence that miRNAs may act locally within developing axons. Kaplan’s group was the first to detect a miRNA in the axonal compartment, miR-338, as early as 2008 [69] and later uncovered its function in the context of circuit development [70]. The authors showed that miR-338 modulates mitochondrial activity in axons of SCG neurons [69,70]. Both loss of function (LOF) and gain of function (GOF) experiments in axons demonstrated that miR-338 regulates the expression of two proteins involved in the mitochondrial oxidative phosphorylation chain: the cytochrome c oxidase subunit COXIV [69] and the ATP synthase ATP5G1 [70]. A clear anticorrelation was observed between the experimentally modulated levels of miR-338 and those of *ATP5G1* and *COXIV* mRNA, which is detected by RT-qPCR. Thus, miR-338 represents an interesting example of a single miRNA coordinating the expression of proteins with related functions. Phenotypically, GOF and LOF experiments modulating miR-338 levels within axons grown in Campenot chambers (Figure 4) revealed that this miRNA acts as a negative regulator of SGC axonal outgrowth [70]. Whether miR-338 mitochondrion regulatory function impacts outgrowth is still unknown.

Additional miRNAs were later found to act within axons to regulate outgrowth. Inhibition of axonal miR-16 in SGC neurons [71] by anti-miR-16 transfection within the axonal side of compartmentalized Campenot chambers elicits axon elongation in culture. miR-16 locally targets the translation initiation factors *eIF2B2* and *eIF4G2* within axons, as revealed by RT-qPCR and Western blotting combined with axonal transfection of double-stranded miRNA mimics or miRNA inhibitors [71] (Figure 4). These transcripts encode two eukaryotic translational initiation factors, and their downregulation in axons, mimicking miRNA silencing, leads to the decreased levels of β-Actin, cytochrome c oxidase subunit 4, mitochondrial (COXIV) and hypoxia-inducible factor-1α (HIF-1α) local translation without affecting mRNA levels. Overall, these observations suggest that miR-16 controls axon outgrowth by impinging on the mRNA translation regulatory machinery itself, shutting down the translation of specific transcripts.

These studies revealed that miRNA clearly operates within axons; however, what modulates their expression, and whether their described targets mediate their action locally, was not uncovered. Building on these earlier discoveries, subsequent work unequivocally unraveled that miRNAs temporally regulate mRNA translation within developing axons.

#### 3.1.2. Modulation of Cytoskeleton Regulators

During axon development, axonal mRNAs are kept silent until exposure to external stimuli leads to their de-repression and the concomitant burst of protein synthesis [72]. miRNAs have been reported to play a pivotal role in the switch from an untranslated to a translated state to fuel axon outgrowth [52,73] and direct steering [41]. In all three cases, an miRNA keeps a specific mRNA in a silent state until environmental exposure releases this miRNA-mediated repression and induces local protein synthesis.

Using exogenous reporter constructs, Dajas-Bailador and colleagues have shown that miR-9 keeps *Map1b* transcript dormant until BDNF stimulation induces its translation [73]. Specifically, endogenous miR-9 localizes in the soma, dendrites, and axons of cortical neurons but represses *Map1b* selectively in distal axons. Endogenous axonal levels of miR-9 assessed by in situ hybridization (ISH) are affected by BDNF exposure: short stimulation with BDNF (10 ng/mL for 2 h) decreases miR-9 expression in axons, while prolonged stimulation of more concentrated BDNF (100 ng/mL for 48 h) increases miR-9 levels in distal axon. The rise in MAP1B levels upon BDNF short stimulation promotes axonal outgrowth, while its decrease upon prolonged exposure of the same cue induced branch formation. Similarly, target protector (Figure 4) designed to disrupt miR-9 access to its MRE on MAP1B elicits an increase in axonal length and a decrease in branch formation in vivo [73]. Overall, these data provided the first evidence of miRNA as key regulatory molecules of mRNA translation acting at a specific time (i.e., after BDNF exposure) and in a precise space (i.e., locally in axon).

A second example of the lifting of miRNA-mediated inhibition in response to extracellular cues comes from the finding that exposure to nerve growth factor (NGF) leads to de-repression of endogenous *Map1b* and *Calm1* mRNAs within axons by counteracting a constitutive repression by miR-181d. This NGF-induced de-repression promotes axon elongation of rodent DRG neurons in cultures. The direct inhibitory role of miR-181d in this regulatory circuit is demonstrated by its abrogation following the delivery of target protectors to the axonal side of a microfluidic device (Figure 4), which is accompanied by increased axonal levels of MAP1B and CALM1 protein [52]. Additionally, both effects of NGF stimulation, increased MAP1B and CALM1 protein levels and enhanced axonal elongation, are blocked by the axonal delivery of miR-181d mimics. As shown in dendrites [74], FMRP might cooperate with a miRNA to locally silence specific mRNAs. FMRP, miR-181d, *Map1b,* and *Calm1* mRNA coprecipitate in neurons, suggesting that this RBP forms regulatory granules with this miRNA and its targets. In addition, the axonal transfection of FMRP inhibitors attenuate the effect of NGF on axonal elongation and block the MAP1B and CALM1 protein synthesis similarly to the effects observed for miR-181d mimics above, reinforcing the notion of RNA binding protein (RBP)/miRNA cooperation [52]. In PC12 cells, NGF induces the dissociation of *Map1b* and *Calm1* from FMRP granules, but not of miR-181d, which remains associated to FMRP [52]. Whether this mechanism also occurs in axons and how the dissociation is mediated remain an intriguing open question. Together, these data illustrate a complex interplay between an RBP and miRNA in modulating cue-induced mRNA translation.

Lastly, miR-182, the most abundant miRNA in *Xenopus laevis* retinal ganglion cell (RGC) axons, silences Cofilin-1 specifically in growth cones and is essential for Slit-2-induced Cofilin-1 translation burst and growth cone steering [41]. Using quantitative immunofluorescence and a kaede-Cofilin1-3′UTR fluorescent reporter construct (Figure 4), we have shown that miR-182 knockdown results in increased Cofilin-1 protein levels in the absence of Slit-2 and abolishes the rapid (5–10 min) Slit-2-induced burst in Cofilin-1 protein synthesis and growth cone steering ex vivo. This suggests that miR-182 impinges on Slit-2-induced protein synthesis and axon turning. Isolated axons were employed here (Figure 4), excluding any possible contribution of trafficked molecules from the soma, thereby unequivocally confirming a local effect. Using a reporter sensor (Figure 4), miR-182 was shown to be exclusively active in axons, where it accumulates, but not in the soma, and to be functionally inactivated by Slit-2 with no changes in its abundance. At the organismal level, miR-182 is critical for accurate axon targeting. MiR-182 downregulation results in aberrant axonal projections specifically within the target area where Slit-2 is expressed, which is a phenotype similar to Slit-2 loss of function within the brain. Overall, this study was the first to unravel the key role of an axonal miRNA in modulating the fast cue-induced mRNA translation in growth cone steering [41].

So far, we reported three miRNA-mediated mechanisms: miR-9 silences Map1B until BDNF application [73], miR-181d silences Map1B and Calm1 until NGF exposure [52], and miR-182 silences Cofilin-1 until Slit-2 stimulation [41]. All these three mechanisms share some important commonalities: (i) the miRNA locally keeps its target in a silent state until the axon is exposed to an external stimulus, (ii) miRNA-regulated transcripts are related to cytoskeletal-remodeling components (e.g., MAP1B and Cofilin-1), (iii) the miRNAs mediate translational repression but not target degradation. These observations suggest a first possible common regulatory mechanism mediated by miRNA to foster axon development. In unstimulated basal conditions, several miRNAs keep different sets of axonal mRNAs dormant in a confined neuronal compartment. Upon stimulation, a subset of miRNAs is selectively inactivated, depending on the external signal, thereby releasing their translational repression. The selected repertoire of de-repressed axonal mRNAs are locally translated, while other mRNAs remain silenced possibly by different sets of miRNAs. The burst of regionalized protein synthesis modulates the cytoskeleton, which in turn regulates the axon behavior in response to the sensed cue (i.e., stimulated outgrowth, directional steering toward or far from the cue) (Figure 5).

How exactly these miRNAs are released from the target upon cue exposure remains to be elucidated. Nevertheless, these findings clearly show a local role of miRNAs in axons to ensure a rapid translation of selected mRNAs locally and on demand, ultimately ensuring the exquisite accuracy required for the different phases of axon development.

#### 3.1.3. Regulation of Receptors to Cues

The correct spatiotemporal expression of guidance receptors is essential for guiding axons precisely at specific choice points en route. For years, the local protein synthesis of guidance receptors has been speculated to occur in order to modulate cue sensitivity directly within axons. The first such receptor shown to be locally synthesized in axons was ephrin type-A receptor 2 (EPHA2) in distal commissural axons (CAs) at the midline through reporter constructs [77]. CAs project toward the midline and cross it, passing from one side of the central nervous system (CNS) to the other. The midline is an important intermediate target, as axons have to reach it but then continue their journey without being trapped at this critical location. For this reason, axons change their sensitivity to midline cues, switching from attraction to repulsion. For instance, pre-crossing axons are sensitive to the attractive cue Netrin-1 and not sensitive to the repulsive cue Slit-2, while the post-crossing axons behave in an opposite fashion.

In this particular context of axonal development, a new miRNA-mediated regulatory mechanism was recently discovered [75]. This study revealed that the regulation of Robo1 expression within crossing axons is essential to modulate their responsiveness to Slit-2 [78]. The authors examined the possibility that local miRNAs are involved in this process using quantitative fluorescence of a photoconvertible Robo1-3′UTR translation reporter in live isolated chick commissural axons following neural tube electroporation of miRNA mimics or inhibitors [75]. These experiments show that miR-92 targets Robo1 mRNA in the axonal compartment, thereby dampening sensitivity to Slit-2 in pre-crossing CAs. In post-crossing CAs, miRNA-mediated inhibition is relieved (by an unknown mechanism) eliciting Robo1 local expression and Slit-2 sensitivity [75] (Figure 5). In pre-crossing commissural neurons, miR-92 is present in axons and growth cones, while in post-crossing neurons, its levels drop with the associated increase of Robo1 protein expression. To assess miR-92 function in midline crossing in vivo, miR-92 was either depleted by sponges (Figure 4) electroporated to the brain, or miR-92 binding to its Robo1 mRNA target was abolished through the use of a target protector. In both cases, most commissural axons stalled at the floor plate, while a subset misprojected, extending far from the midline. This suggests that the regulation of Robo1 expression by miR-92 is important for midline crossing at the organismal level. Overall, this study unraveled the key role of miRNAs in locally modulating cue receptor expression to guide axons through an essential choice point.

#### 3.1.4. Regulation of Signaling Pathway

The communication between the axon and the soma is critical for axon development, as external environment signals sensed by distal axons are transduced into changes in gene expression at the cell body. Seminal studies have revealed that such a mechanism relies on the signal-induced retrograde transport of locally translated proteins to the nucleus to elicit transcription [8,79,80,81]. A recent study demonstrates that miRNAs play a crucial role in this process [76]. When miR-26a silencing of glycogen synthase kinase-3 beta (*Gsk3β*) transcripts is relieved in axons, newly synthesized GSK3β is transported to the soma, where it impacts axon polarization and outgrowth [76]. The localized role of miR-26a in axons was studied in mice primary cortical neurons using microfluidic devices. miR-26a inhibitor or target protector against miR-26a MRE within *Gsk3β* 3′UTR reduces axonal outgrowth and the number of polarized neurons exclusively when transfected to the axonal side but not to the somatic side, suggesting a local role for miR-26 in this process. Intriguingly, the authors provide several lines of evidence suggesting that the miR-26a-induced newly synthesized GSK3β protein is retrogradely transported from the axon to the cell body for action. First, miR-26a inhibition within the axonal compartment increases GSK3β protein level in both axon and soma, while miR-26a inhibition at the soma does not change GSK3β expression. Second, GSK3β inhibitor rescues miR-26 inhibitor-mediated axon growth phenotype both when delivered to the axonal and to the somatic side [76]. Third, destabilizing microtubules with nocodazole after miR-26 inhibition in axons confines the GSK3β protein surge to the axon compartment, indicating that the previously observed increased levels in the soma entirely depends on the retrograde transport of newly synthesized GSK3β toward the cell body. The signal relieving the miR-26-mediated repression of *Gsk3β* mRNA remains to be identified. Nevertheless, the axonal synthesis of GSK3β after miR-26 de-repression and GSK3β relocalization to the soma is clearly an important regulatory mechanism for an appropriate neuronal polarization and outgrowth (Figure 5). This discovery highlights a new miRNA-mediated molecular mechanism to regulate the spatiotemporal protein expression at the axonal level in the context of axon-soma communication.

In a similar vein, miR-19a locally regulates within axons the translation of another signaling molecule, phosphatase and tensin homolog (PTEN), which is a critical inhibitor of the phosphatidylinositol 3-kinase (PI3K)/AKT pathway [82]. Modulating miR-19a levels locally in microfluidic chambers has revealed that axonal miR-19a promotes axon elongation. *PTEN* is a direct target of miR-19a in axons of rat cortical neurons. Accordingly, inhibiting miR-19a in distal axons induces an increase in PTEN protein expression and a decrease in mTOR (mammalian target of rapamycin) and GSK-3β phosphorylation, thus inactivating the PI3K signaling pathway (assessed by Western blotting) in axons but not in cell bodies. In whole neurons, PTEN over-expression rescues the abnormal outgrowth phenotype elicited by miR-19a GOF, while dampening PTEN levels by RNAi prevents the cellular abnormalities induced by miR-19a LOF. This indicates that PTEN is an effector of miR-19a in promoting outgrowth [82]. Whether PTEN is a local mediator of axonal miR-19a is still unknown. Overall, these data indicate that axonal miR-19a may impact one of the major signal transduction pathways to promote axonal development.

### 3.2. Axonal Stimulation Triggers miRNAs-Mediated mRNA Silencing

The studies reviewed so far exemplify a general mechanism whereby bursts of mRNA translation occur upon alleviation of a basal level of inhibition carried out by specific miRNA within the axon. Our recent work shows that the opposite mechanism also takes place within the axonal compartment: an miRNA is activated upon cue exposure, leading to the silencing of basally translated transcripts, ultimately promoting change in growth cone behavior [45]. We found that inactive pre-miRNAs are shipped to the growth cone, where they are stored. An environmental stimulus induces their local processing into active, newly generated miRNAs (NG-miRNAs), which then proceed to silence their target transcripts. Conceptually, as localized mRNAs represent a pool of inactive protein, pre-miRNAs similarly represent a pool of inactive miRNAs. Some corollaries of such mechanism based upon the sub-localization of pre-miRNAs that can mature locally are manifestly advantageous in the context of axon physiology: (i) it provides a ready-to-use inactive pool of regulatory molecule in a confined space that can be activated at the right time, (ii) it prevents miRNAs from inhibiting target mRNAs during transport, before reaching the cellular domain where such inhibition is needed, and (iii) it restrict the number of potential mRNA targets to only those present in that specific sub-compartment.

We found that Semaphorin 3A (Sema3A) induces pre-miR-181a-1/a-2 local processing into newly generated mature miR-181a-5p and miR-181a-3p in a cue- and miRNA-specific manner in isolated axons by RT-qPCR [45]. We identified tubulin isoform beta-III (*TUBB3*), the most dynamic microtubule isoform [83], as an miR-181-5p target through axonal RNA-seq and in silico target analysis. Fluorescence recovery after photobleaching (FRAP) experiments using Venus-TUBB3-3′UTR reporter [84] (Figure 4) revealed that TUBB3 is locally translated in growth cones at the basal level and is downregulated upon Sema3A exposure [45]. We further observed that Sema3A-mediated *TUBB3* mRNA translation downregulation is carried out by NG-miR-181a-5p, as blocking its maturation in isolated axons, or preventing its action by mutating miR-181a-5p MRE in *TUBB3* abolishes the ability of Sema3A to dampen TUBB3 expression [45]. Puro-PLA, a technique that allows the visualization of translational events [85], confirmed these results on endogenous TUBB3 levels in isolated axons. This RNA-based pathway appears to be crucial for Sema3A-mediated effects on developing axons: blocking pre-miR-181a-1 processing abolished the Sema3A-induced growth cone collapse response, while a simultaneous block of *TUBB3* translation restored it. These results suggest that specific cytoskeleton components are translated during unstimulated, basal axonal outgrowth on laminin substrate and are locally downregulated by a repellent cue to induce growth cone steering.

This novel RNA-based regulatory pathway is critical at the organismal level for neural circuit assembly and animal behavior. Indeed, axonal miRNA-mediated TUBB3 regulation takes place in vivo within the axon targeting region, the optic tectum, as shown using in vivo single-axon FRAP. RGC axons depleted in the miR-181 family aberrantly project within the optic tectum, their target brain area, and these morphant embryos display impaired vision. Dicer is the key endonuclease responsible for the processing of pre-miRNA into mature miRNA. The detection of Dicer in mouse RGC axons during the period of axon targeting [45] suggests that a similar mechanism may be at play during mammalian visual system development as well.

Support for a cue-induced miRNA activation and translational repression comes from another study. Chondroitin sulfate proteoglycans (CSPGs) are extracellular molecules that inhibit the axonal outgrowth of rat cortical neurons and modulate the miRNome within axons [86]. The application of CSPGs specifically to axons grown in a microfluidic chamber induces axonal miR-29c upregulation and Integrin-β1 (ITGB1) downregulation [86]. Axonal miR-29c itself modulates levels of ITGB1 and mediates CSPGs effect on outgrowth regulation. It would be interesting to understand how miR-29c is activated by CSPGs, whether axonal pre-miR-29 processing is involved and whether miR-29c acts locally to regulate the local translation of ITGB1.

### 3.3. Stage-Dependent miRNAs-Mediated mRNA Silencing

Most studies exploring the local roles of miRNAs in mRNA translation have examined whether miRNAs activity could be acutely modulated by cue exposure. However, the composition of the cellular miRNA pool is known to also change gradually over time in neural tissues as development proceeds [40,75,87,88,89,90]. MiRNA can induce silencing when it is present in the cell at adequate stoichiometry with respect to its targets. miR-132 expression peaks in the period of maximum axon outgrowth in mouse DRG neurons in vivo [40]. GOF and LOF approaches have revealed that this axon-enriched miRNA promotes axon elongation in culture. MiR-132 locally regulates the translation within axons of p120RasGAP (Rasa1), which is a Ras GTPase activator [40]. Rasa1 acts as a suppressor of the function of Ras, which is an important component of the MAPK/ERK (Mitogen-activated protein kinase/Extracellular signal-regulated kinase) signaling pathway known to promote axon development [91,92,93] and an activator of PI3K [94]. MiR-132 local inhibition in severed axons increases Rasa1 protein level (Western blot) without affecting the mRNA (RT-qPCR), suggesting that this miRNA regulates the local translation of Rasa1. Evidence suggests that Rasa1 mediates miR-132 action on axon outgrowth, as the transfection of a target protector masking miR-132 MRE within Rasa1 3′UTR restored miR-132 mimics-induced impaired axon outgrowth in Neuro2A cells. This suggests that miR-132 promotes axon outgrowth by downregulating an inhibitor of the MAPK/ERK and PI3K signaling pathways. The Rasa1 mRNA pattern of expression mirrors that of miR-132, as it progressively decreases from E12.5 to E17.5, when miR-132 increases. Together, these data suggest that an axonal miRNA intervenes at specific steps of neuronal circuit assembly, when its expression level is high.

## 4. Conclusions and Perspectives

A developing axon displays a large degree of autonomy and relies in part on local mRNA translation to promote its growth, steering, targeting, branching, and synaptogenesis, which are all critical steps in neuronal circuit development. Local translation is a process that is highly regulated in space and time, and axonal miRNAs are emerging as critical components of the mRNA regulatory network. The miRNA repertoire in axons is complex, and initial studies point to an endosome-based mechanism of transport underlying the proper translocation of these small RNAs to the axonal domains where they exert their function. Our current understanding of miRNA-mediated mechanisms in neural circuit assembly points to two key regulatory pathways likely to work in parallel. On the one hand, a cue-induced mRNA de-repression via miRNA inactivation triggers a burst of translation of specific transcripts. On the other hand, a cue-induced mRNA repression via NG-miRNA activation induces translation inhibition. Together, the two mechanisms would contribute to regulate growth cone behavior in response to external signals by fine tuning in space and time the translation of mRNAs to ultimately remodel the cytoskeleton (Figure 6). This RNA-based pathway may act alongside the classic protein-based regulatory pathway to confer a higher degree of regulatory potential that supports the precision required in neural circuit development.

Research on the local roles of miRNAs in axons has been intense in the last 10 years, yet many areas remain unexplored. It is still largely unclear how miRNAs are specifically targeted to the axonal compartment and within the axon subregions. Does a zip code exist on such short molecules similar to mRNAs, as shown for miR-29b for nuclear import [95]? Are miRNAs mostly transported as pre-miRNA and processed locally? How widespread is endosome-mediated transport as a mode of transport for miRNAs and RNAs in general? Since the endosome acts as local translational hubs in axons [17], is this organelle crucial in miRNA-regulated translational events in response to external stimuli? Could it spatially restrict the access of newly generated miRNAs to a few targets? Numerous studies have revealed that miRNAs activity is modulated by extrinsic signals. Yet, it is unclear how miRNAs are activated or inactivated upon cue exposure. miRNA inactivation may occur through target-mediated degradation in neurons [96,97]. However two studies have shown that this is not the case, since miRNA levels are not altered [41,52]. MiRNA inactivation could be triggered by miRNA modification [98] or displacement by a molecule competing for the same motif such as an RBP. Circular RNAs (circRNAs) could also be involved, as they can act as miRNA sponges [99,100,101], and it is possible to envision that their levels are increased upon stimulation, leading to the sponging/inactivation of miRNAs. We have uncovered a novel mechanism whereby miRNA activation relies on the local processing of the miRNA precursor [45]. However, other mechanisms may exist.

Clearly, our understanding of miRNA-mediated mechanisms in axon development during circuit assembly remains partial, and further investigations are needed to attain a comprehensive description of how these small non-coding RNAs specifically exert their function. Research in this area will not only help dissecting important molecular mechanisms occurring in neuronal development, but it is assured to also impact research in axon regeneration and in neurological pathologies. Indeed, axon regeneration can be considered a recapitulation of developmental growth. Axonal translation is a crucial mechanism to promote regeneration [102,103], and not surprisingly, miRNAs act as regulators in the regenerative response of injured axons both in the peripheral and central nervous system [104]. In addition, miRNA dysregulation has been linked with neurodevelopmental and neurodegenerative diseases [105]. However, local roles of miRNA in axons in these contexts are largely unexplored. All in all, axonal miRNAs appear to be key regulators throughout the brain lifespan, with potential implication in regenerative processes and in pathological conditions. This characteristic makes them prime targets for therapeutic intervention, and improving our understanding of miRNA-mediated mechanisms might potentially help with treating neuronal trauma and nerve injury in addition to neurodevelopmental and neurodegenerative disorders.

## Figures and Tables

**Figure 1 ijms-21-08726-f001:**
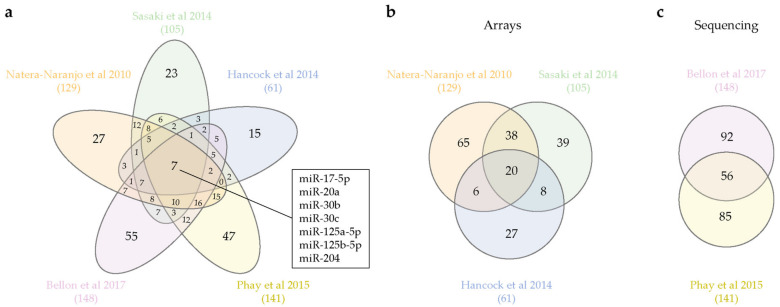
Axonal miRNAs shared among datasets. (**a**) Venn diagram of all axonal miRNAs profiling. Seven miRNAs are shared by all datasets, and names are reported in the box. (**b**) Venn diagram of profiling obtained by miRNAs microarray [38] or TaqMan RT-qPCR arrays [39,40]. (**c**) Venn diagram of miRNAs shared between sequencing datasets. Total number of axonal miRNAs are reported in parenthesis below the references. Venn diagrams are obtained with an open access software [47].

**Figure 2 ijms-21-08726-f002:**
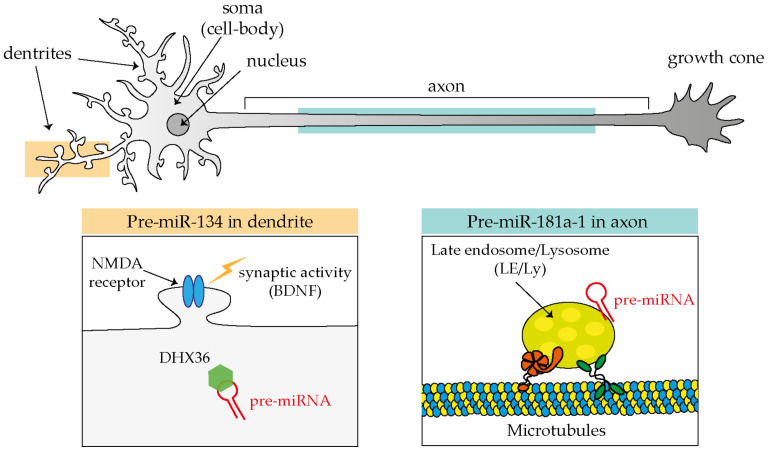
MiRNAs transport in neuronal compartments as inactive pre-miRNAs molecules.

**Figure 3 ijms-21-08726-f003:**
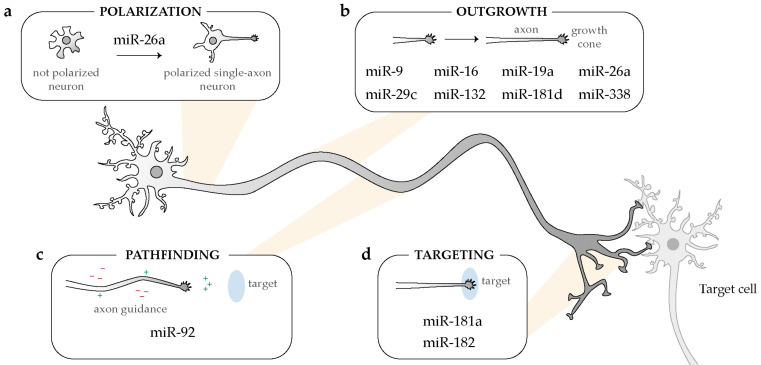
Overview of the broad impact of intra-axonal acting-miRNAs on several steps of neuronal development: polarization (**a**), outgrowth (**b**), pathfinding (**c**), and targeting (**d**).

**Figure 4 ijms-21-08726-f004:**
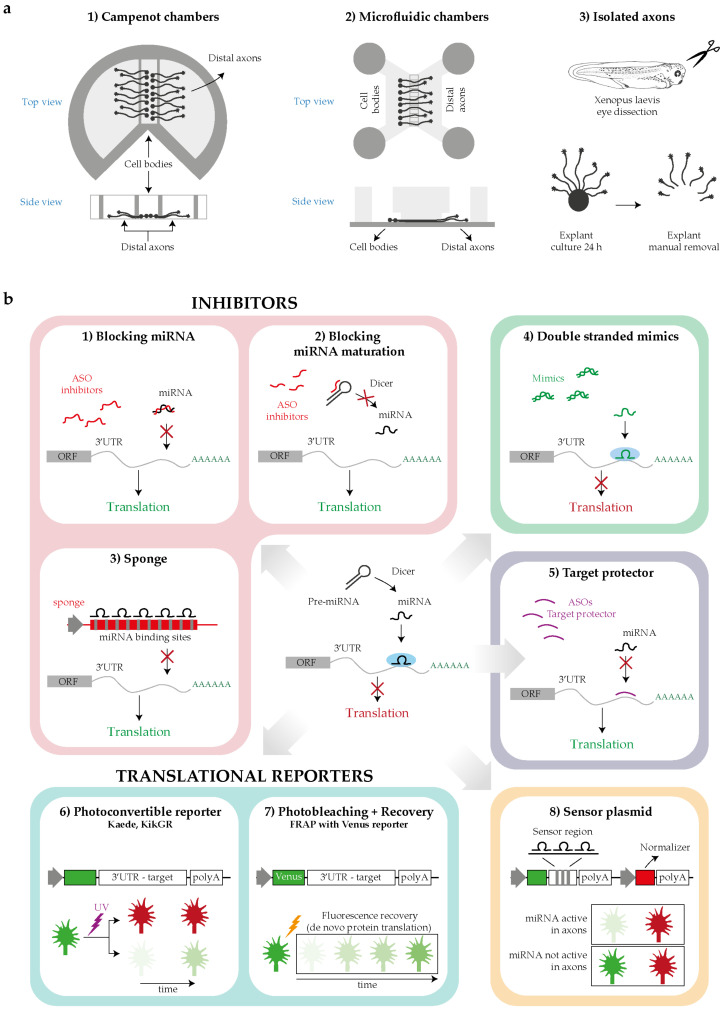
Tools for investigating the intra-axonal roles of miRNAs. (**a**) Schematics for distal axons isolation: Campenot chambers, microfluidic devices or manual removal of the explants in case of organoculture. (**b**) Techniques for studying miRNA function: by loss of function (LOF) approach blocking miRNA function (1–3), by gain of function (GOF) approach increasing miRNAs level with mimics (4); by blocking miRNA target recognition, masking the miRNA-responsive element (MREs) with a target protector (5); by studying with reporters the translational events potentially regulated by the miRNA under study (6,7); by measuring the miRNA activity using sensor plasmids (8). Abbreviations: ORF, open reading frame; ASOs, antisense oligonucleotides.

**Figure 5 ijms-21-08726-f005:**
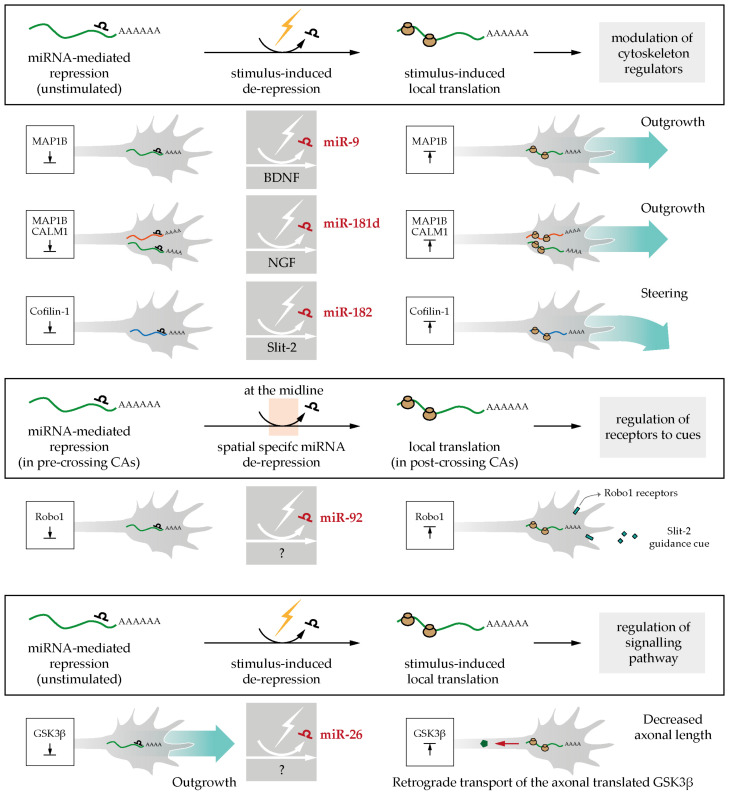
Axonal stimulations relieve miRNAs-mediated mRNAs silencing: role in axonal pathfinding. (i) brain-derived neurotrophic factor (BDNF), nerve growth factor (NGF) or Slit-2 stimulation induce miRNA de-repression and gate the translation of cytoskeleton regulators (MAP1B, CALM1, Cofilin-1) thereby regulating outgrowth and steering [41,52,73]; (ii) miR-92 keeps Robo1 silent in pre-crossing commissural axons (CAs), and miRNA de-repression at the midline triggers Robo1 translation and concomitant axon sensitivity to cue [75]; (iii) miR-26 de-repression in axons elicits glycogen synthase kinase-3 beta (GSK3β) local synthesis and this leads to the decrease in axonal outgrowth [76].

**Figure 6 ijms-21-08726-f006:**
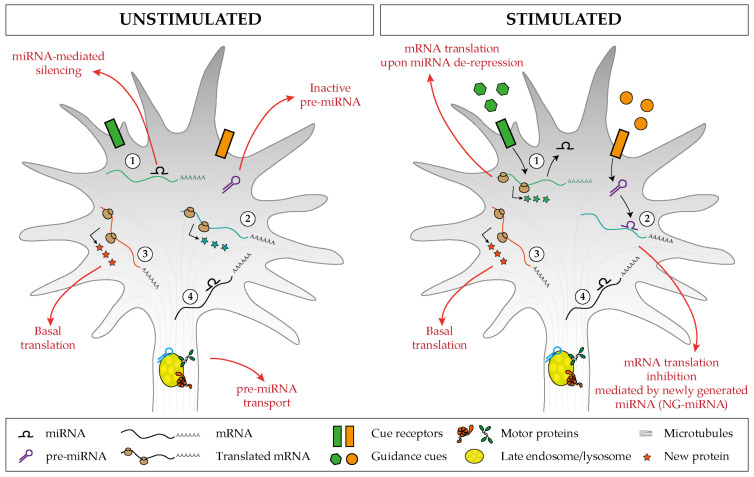
Proposed model of miRNA-mediated regulation of basal and cue-induced translation. (1) Specific miRNAs keep a subset of mRNAs in an untranslated state, but upon stimulation, the miRNA-mediated silencing is relieved, and this induces a burst of mRNA translation. (2) Inactive pre-miRNAs are transported and stored within the axonal compartment (left), and upon cue exposure (right), distinct pre-miRNAs are locally processed and the associated newly generated miRNAs (NG-miRNAs) inhibit the basal translation of given transcripts. Regardless of the stimulus status, (3) specific transcripts undergo basal translation and (4) others are maintained in an untranslated silent state.

**Table 1 ijms-21-08726-t001:** Axonal miRNAs.

Species	Neuronal Type	Axonal miRNAs	Axon Enriched miRNAs	Technique	Ref
Rat	SGC sympathetic neuron	130	miR-15b, -16, -204, -221	miRNA microarray kit	[38]
Mouse	Cortical neuron	105	miR-134, -181a-1-3p, -361, -532, -685, -709, -720	RT-qPCR array	[39]
Mouse	DRG primary sensory neuron	61	miR-24, -132, -138, -191	RT-qPCR array	[40]
Mouse	Motor neurons	not defined	let-141, -200b, miR-15a*, -18a, -21*, -29b, -29c, -30e, -33, 101a, -142-5p, 148a, -153, -154*, -183, -184, -193, 196a*, -200a-5p, -200a-3p, -200c, -301a, -322, -338-3p, -363, -375, -376a, -486, -494, -497, -542-3p, -669d, -708*, -879	HiSeq 2500 sequencing	[46]
Rat	SN axoplasm	141	not investigated	HiSeq 2500 sequencing	[42]
Xenopus laevis	RGC	148	not investigated	MiSeq sequencing	[41]

Abbreviations: SGC, superior cervical ganglia; DRG, dorsal root ganglion; SN, sciatic nerve; RGC; retinal ganglion cells.

**Table 2 ijms-21-08726-t002:** Axonal pre-miRNAs.

Species	Neuron Type	Pre-miRNAs Presence	Technique	Pre-miRNAs Characterization *	Technique	Ref
Mouse	DRG	pre-miR-132	RT-qPCR	X	X	[40]
Rat	DRG	pre-miR-16, -23a, -24-1,-25, -26a, -103-1, -125b-1,-127, -134, -138-2, -185,-221, -329, -382, -433, -541	PCR	pre-miR-25, -433	ISH	[43]
Rat	SCG	pre-miR-134, -185, -204, -338	RT-qPCR	pre-miR-338	RT-qPCR from SCG axon mitochondrial fractions, imaging of exogenous labeled pre-miR-338	[44]
Xenopus laevis	RGC	pre-let-7f, -7g, pre-miR-9a-1,-9b, -9-3, -16b, -18, -19b, -26-2,-27a, -29d, -96, -100, -103-1,-124-5, -126, -129-1, -130a,-133c, -139, -140, -143, -145,-148b, -153-1, **-181a-1, -181a-2,-182,** -199a, -199b, -204-1, -208,-212, -214, -222, -223, -301-2,-375, -428, -455-1, -703, -1306	MiSeq sequencing, **PCR, RT-qPCR**	pre-miR-181a-1	Live imaging of endogenous (MB-labeled) and exogenous (cy3-labeled) pre-miR-181a-1	[45]

The subset of axonal pre-miRNAs further studied by PCR and RT-PCR are shown in bold [45]. * Axonal presence validation [43,44,45] or characterization of pre-miRNA localization [44], trafficking [45]. Abbreviations: DRG, dorsal root ganglion; SGC, superior cervical ganglia; RGC, retinal ganglion cells; ISH, in situ hybridization; MB, molecular beacon.

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
