# Peer review of "In the Right Place at the Right Time: miRNAs as Key Regulators in Developing Axons"

_ijms, 2020, doi:10.3390/ijms21228726_

Round 1
Reviewer 1 Report
The review by Corradi and Baudet provides a comprehensive, up to date and well-written summary of the growing field of miRNAs in axonal development. This is a timely addition to the literature, and one that contributes an in-depth understanding and description of the molecular mechanisms at play. As such, I believe that it merits publication and I have only made some suggestions to improve the final version of the manuscript.
Section 1.
- Line 35-36: Although not the main scope of the review, it would be good to add some reference to the more historical papers when mentioning that “mRNA translation in axons has been clarified after decades of investigation”, such as those by Koenig, Guiditta, et al.
- Line 35-50: This paragraph makes a good summary of the latest findings, however one area that could also be mentioned is that beyond its initial suggestion as a developmental mechanism, local axon translation is starting to be accepted as a cellular process of importance throughout the lifespan of a neuron/brain. For example, evidence of mRNA and translation machinery in adult axons could be added.
- Line 54: I would say that in terms of sncRNAs it is true that miRNAs are the “overwhelming” majority reported in the literature, but this sentence could include other sncRNAs as potential regulators with growing interest (i.e. piwi-RNAs). In this same paragraph I would also suggest the addition of some references to guide the non-miRNA expert.
- Line 57: Sentence starting with “Free from the constraints…” is rather long and should be edited.
Section 2.
- Line 102-105: This meta-analysis approach is a welcomed addition to the growing field of axonal miRNAs, even if the paucity of published datasets means different species have to be compared. However, the NGS analysis would benefit from the incorporation or mention of human data from the Rotem et al. paper (ALS along the axons-Expression of coding and noncoding RNA differs in axons of ALS models; Scientific Reports, 2017).
It is good to see the separate discussion of both “axonal lists” and “enrichment analysis”, as sometimes these are confused in the literature, with the latter having the added caveat that sometimes normalisation pipelines might not be comparable and/or adequate.
Section 3 and 4.
- These sections provide a very clear narrative and detailed description of the miRNA mechanisms in the axon, with great diagrams that provide a very good complement to the text. One thing to suggest though is that legends of Figures 4 and 5 in particular could be expanded as they are currently rather brief. In Figure 5, the growth cones could be increased in size to allow “bigger” mechanistic diagrams inside, which in some cases are not easily visualized in the current proof version.
- Line 316: I would suggest changing miRNAs-mediated for miRNA-mediated.
- Line 334, Section 3.1.4 Regulation of signalling pathway. I have some doubts about the title proposed for this section, which is rather vague. Moreover, although the GSK-PTEN-Pi3K axis is indeed a key signalling pathway involved in growth, it is also linked to cytoskeleton mechanisms, so one can argue that the miRNAs described here (miR-19a and miR-26a) could also be part of section 3.1.2. Instead, an alternative title for this section might be “Regulation of retrograde communication from the axon” or similar, using miR-26a as the miRNA example that builds on the seminal studies on local translation and retrograde transport by the labs of Twiss, Hengst, Jaffrey and others, which is how the section currently starts. In this new structure, miR-19a regulation of PTEN and the GSK link of miR-26a could be added to a renamed 3.1.2 Modulation of Cytoskeleton regulators and signalling pathways. I am sure the authors must have thought extensively about the current structure, so this is a suggestion more than a request, but I believe it better reflects the overall mechanistic narrative presented.
- Section 4. This is a nicely written section that conveys the main points well. I would perhaps add a comment that the developmental role of miRNAs in axon (main focus of this review), is likely to be expanded into adult neuronal networks in the future, with growing number of studies addressing their potential role in neurodegenerative diseases.
Reviewer 2 Report
This review describes the role of axonal miRNAs, including their transport along the axon, and seems particularly well-suited as a thorough survey or introduction to the field. The illustrations appear especially well-constructed. The text is logically organized and the tables are informative. Minor quibbles include some potentially messy verbiage, e.g. line 193, “miRNAs mechanism of actions will be discussed next” seems redundant; line 236 uses “derepression” while line 450 uses “de-repression.” Additionally, line 57, “Free from…” would benefit from reference(s) in regards miRNAs “rapidly evolving” and “acting through imperfect matches to their target sequence.” Nevertheless, on the whole, this manuscript enhances the scientific discourse and appears worthy of publication.
Reviewer 3 Report
In order to become part of the extremely complex network of the nervous system, neurons must undergo a series of developmental events including the formation, extension and navigation of axons -through a strict guidance program- towards specific postsynaptic targets that axons must in turn recognize appropriately and connect to in a process involving extensive protein dynamics. Most of these incredibly intricate phenomena occur far from the neuronal soma indicating that axons must acquire some degree of autonomy controlling the translation of locally available mRNA transcript to fine-tune their specific functions by instructing protein production in situ. Over the last decade it has become increasingly evident that miRNAs are key players regulating local translational responses in axons holding an impressive translational potential for their therapeutic use to treat neurodegenerative disorders. Nevertheless, the complex traits of miRNA biology, sub-cellular sorting and regulation in neuronal systems have been only superficially grasped.
In their review, Corradi and Baudet offer a concise overview of relevant studies revealing different aspects of miRNA function in developing axons including regulation of critical effector molecules for axonal behavior as well as diverse miRNA roles in the development of axons including their formation, extension, branching and guidance.
The manuscript is well written and organized. The images are appropriate to help conveying the message of the sections and the literature is up to date. I envision that this review by Corradi and Baudet will be well received by the community.
A few comments and edits:
>In the section conclusions and perspectives, it is discussed the potential of miRNAs as targets for the treatment of neurodevelopmental and neurodegenerative diseases. I would like to point out that there’s ample evidence in the literature indicating mediation of miRNAs in the regenerative response of injured axons both in the peripheral and central nervous system. Since axon regenerations is considered a recapitulation of developmental growth, it would be worth mentioning, at least in this final section the therapeutic potential of miRNAs as targets to treat neuronal trauma and nerve injury in addition to neurodegenerative disorders.
> It would be great to have a visual input (figure) of the different molecular tools mentioned across the text i.e. UTR-reporters; target protectors; miRNA inhibitors; double-stranded mimics; inactivation of miRNA by “sponges”; etc. This addition could greatly help readers to better grasp the virtues of the methodologies introduced by the author.
Line 28: Maybe synaptic partners is a better suited term than "developing dendritic spines" since synapses are not necessarily always established at dendritic spines (typically only 80-90%).
Line 45-46: [2,3] recent studies revealed additional focal hotspots……
Line 251: translation acting at a specific…….
Line 253: A second “example” instead of “illustration”…..
Line 286: “All these three mechanisms…” Maybe re-introduce the mechanisms for more clarity.
Lines 407-410: Sentence is too long and difficult to digest. Perhaps make it short or rephrase it.
Lines 428-431: Is not clear to which references the miRNA-132/Rasa 1 study belongs to.
